# Chemical Vapour Deposition of Scandia-Stabilised Zirconia Layers on Tubular Substrates at Low Temperatures

**DOI:** 10.3390/ma15062120

**Published:** 2022-03-14

**Authors:** Agata Sawka

**Affiliations:** Department of Physical Chemistry and Modelling, Faculty of Materials Science and Ceramics, AGH University of Science and Technology, 30 Mickiewicza Av., 30-059 Krakow, Poland; asawka@agh.edu.pl

**Keywords:** Sc_2_O_3_-stabilised ZrO_2_ layers, electrolyte, tubular SOFC, MOCVD

## Abstract

The paper presents results of investigation on synthesis of non-porous ZrO_2_-Sc_2_O_2_ layers on tubular substrates by MOCVD (metalorganic chemical vapor deposition) method using Sc(tmhd)_3_ (Tris(2,2,6,6-tetramethyl-3,5-heptanedionato)scandium(III), 99%) and Zr(tmhd)_4_ (Tetrakis(2,2,6,6-tetramethyl-3,5-heptanedionato)zirconium)(IV), 99.9+%) as basic reactants. The molar content of Sc(tmhd)_3_ in the gas mixture was as follows: 14, 28%. The synthesis temperature was in the range of 600–700 °C. The value of extended Gr_x_/Re_x_^2^ expression (Gr-Grashof number, Re-Reynolds number and x-the distance from the gas inflow point) was less than 0.01. The layers were deposited under reduced pressure or close to atmospheric pressure. The layers obtained were tested using scanning electron microscope (SEM) with an energy dispersive X-ray spectroscope (EDS) microanalyzer, X-ray diffractometer and UV-Vis spectrophotometer. The layers deposited were non-porous, amorphous or nanocrystalline with controlled chemical composition. The layers synthesized at 700 °C were nanocrystalline. ZrO_2_-Sc_2_O_3_ layers with 14 mol.% Sc_2_O_3_ content had a rhombohedral structure.

## 1. Introduction

Solid oxide fuel cells (SOFC) have been intensively developed for many years. They offer clean, ecological electricity production. They have beneficial features such as high efficiency, low emission, low noise and ease of modularity. Their disadvantages are high manufacturing and operating temperatures. Their lowering to the temperatures of 600–800 °C should improve their reliability and significantly reduce costs by using less expensive materials for SOFC elements, e.g., [1,2,3,4,5].

SOFC consists of two porous electrodes (anode and cathode), dense electrolyte, and interconnects. O^2−^ ions are transported through electrolyte from cathode (where oxygen is present) to anode (where fuel is present). The free electrons resulting from the reaction at the anode take part in generating electrical current that flows through the external electrical circuit. It should be noted that there are two types of SOFC: planar and tubular. Planar SOFC have been widely investigated for a few decades. They are characterised by simplicity and lower manufacturing costs compared to tubular cells. However, their tubular geometry allows to solve problems with start-up time, thermal cracks and sealing [1,2,3,4,5,6,7,8,9,10,11,12,13]. These properties make them more desirable, and the possibility of their manufacturing and operating at low temperatures should allow for a much wider scope of their application.

The commonly used electrolyte for SOFC is yttria–stabilised zirconia (YSZ). However, its ionic conductivity is the highest at temperature of 1000 °C, e.g., [7,8,9,11,12,14,15]. It is obtained by powder sintering at temperatures of 1450–1500 °C [9,14]. Such high temperatures of its manufacturing and operating are the reason of high cost of these cells and, as mentioned, their limited use. Among zirconia-based materials, scandia-stabilised zirconia (ScSZ) seems to be an interesting alternative to the YSZ used so far, due to its high ionic conductivity already at temperatures of 700–850 °C [16,17]. This is related to the similar magnitudes of the Sc^3+^and Zr^4+^ ion radii. The Sc^3+^ ionic radius (0.87 Å, coordination number 8) is smaller than that of Y^3+^ (1.019 Å) and more similar to that of Zr^4+^ (0.84 Ǻ) [18]. For this reason, it is also difficult to obtain a phase equilibrium state [19]. Due to the mentioned similar sizes of ionic radii of Zr^4+^ and Sc^3+^, many polymorphic phases can be obtained. For this reason, the structure of ZrO_2_-Sc_2_O_3_ also significantly depends on the conditions of its preparation [20,21]. At least 7 phases were identified in Sc_2_O_3_-rich regions of the phase diagram (the range of 0–25 mol.%) [19,22]. According to Badwal et al. [19,22], the regular (cubic) phase occurs at a molar contribution of Sc_2_O_3_ of about 9%. Below this value up to 5 mol.% t’ phase (distorted regular phase) appears. When the molar content of Sc_2_O_3_ is above 10% to 12.5%, then a rhombohedral phase characterised by low ionic conductivity occurs together with the regular phase [19]. In contrast, Agarkov et al. [23] consider that a regular solid solution can be obtained with a Sc_2_O_3_ content in the range of 8–12 mol.%, while the monoclinic, tetragonal, tetragonal t’ and t”, and rhombohedral phases can exist with a molar dopant content of 5 to 12 mol.%. From the point of view of electrical properties, it is preferable if the electrolyte has a regular structure [3,16,17,18,19,20,21,22,23,24,25,26,27,28,29,30,31,32,33,34,35,36,37]. However, at lower temperatures, i.e., below 650 °C, it is transformed into a rhombohedral phase with low ionic conductivity [24,25,26] and possibly a tetragonal one [17,26]. On the other hand, ScSZ with a rhombohedral structure at the ambient temperature, undergoes a transformation into ScSZ with a regular structure at the temperatures of 500–650 °C. This transformation is accompanied by volume shrinkage which leads to the formation of cracks in the material.

Apart from the difficulties in determination of the appropriate amount of dopant in the material there is also the problem of material ageing [22,27,28]. Heating of this electrolyte at temperatures of 800–1000 °C can significantly affect its conductivity. According to Badwal et al. [22], a freshly prepared material containing 7.8 mol.% dopant after annealing at 1000 °C for several days showed a decrease in conductivity of up to approx. 70%. The change in conductivity was very significant already after the first few hours of annealing. The deterioration of conductivity for ScSZ was much faster than for YSZ. However, it should be noted that Haering et al. [28] have not observed a decrease in the ionic conductivity of ScSZ after its annealing at 1000 °C when the molar Sc_2_O_3_ content was above 10%.

To stabilise the cubic structure of ScSZ, other admixtures are additionally introduced in small concentrations (1–2 mol.%). Most often these are admixtures of rare earth elements or their oxides: CeO_2_, Gd_2_O_3_, Sm_2_O_3_, Bi_2_O_3_, Yb_2_O_3_,Eu_2_O_3_ and Al_2_O_3_, e.g., [3,17,18,19,22,24,26,27,28,29,30,31,32,37,38]. It is also believed [25,39,40,41] that this effect can be achieved by providing material with a nanocrystalline microstructure. Addition of dopants may cause their segregation during long-term cell operation and the consequent reduction in its ionic conductivity and mechanical strength [25]. Moreover, it should be noted that the microstructure of the electrolyte has an influence on its ionic conductivity as well as its mechanical strength. The higher ionic conductivity will be exhibited by an electrolyte with a nanocrystalline than with a microcrystalline microstructure [18,42]. For the above reasons, it appears that research work should be concentrated on obtaining of an electrolyte with small grains.

The ScSZ electrolyte is mainly obtained by powder sintering, which requires high temperatures, i.e., about 1400–1450 °C, e.g., [16,17,18,19,20,22,24,25,27,28,29,31,42]. There have also been studies on manufacturing of this electrolyte from slurries [13,42] and in the form of films using, e.g., the sol-gel method [21,33], chemical solution deposition (CSD) [35], atmospheric plasma spraying (APS) [36], and physical vapour deposition (PVD) methods [26,30,34,37]. In the case of obtaining electrolyte from suspensions, or using sol-gel or CSD methods, the material obtained had to be sintered at the final stage of the process at temperatures above 1000 °C in order to eliminate pores. On the other hand, the layers deposited by the APS method showed significant porosity, which excludes the possibility of their using as an electrolyte. However, it is difficult to find information on the preparation of layered ScSZ electrolytes by chemical vapour deposition (CVD) and metalorganic chemical vapour deposition (MOCVD) techniques. In work [32], ScSZ and Al_2_O_3_-doped ScSZ layers were synthesised by the MOCVD method in the presence of plasma. Interesting results were obtained in the work [43], where ScZ layers were deposited on porous substrates using this technique.

It should be noted that all the works cited are related to manufacturing of ScSZ electrolyte for planar SOFC. Considerably less information is available on the preparation of this electrolyte for tubular cells. Few papers [1,2,4,5,6,44,45,46,47,48,49,50,51] demonstrate results of investigation on manufacturing of ScSZ electrolytes for tubular SOFC via “wet chemistry” methods, mainly dip-coating, where the final stage of this process is their sintering at temperatures of 1400–1500 °C. However, there is no information on attempts to obtain electrolytes on tubular substrates using CVD or MOCVD methods. CVD and MOCVD methods allow to obtain non-porous amorphous or polycrystalline layers (nanocrystalline or monocrystalline layers) on flat substrates as well as with complex shapes and large dimensions. These methods are useful for the layer deposition from materials characterised by small diffusion coefficients as ceramic materials (with a large content of covalent binding). Properly conducted CVD and MOCVD processes make it possible in obtaining dense and uniform in thickness layers with controlled chemical composition (in the case of two or more components) and good adhesion to the substrate. The use of more reactive metalorganic compounds (MOCVD) allows to reduce considerably the synthesis temperature. The results of the research conducted so far at the AGH University of Science and Technology (Faculty of Materials Science and Ceramics) in Krakow (Poland) confirm the high usefulness of this method for the synthesis of non-porous ceramic layers on tubular substrates at low temperatures. These results have been published recently in the papers, e.g., [52,53,54,55,56,57].

The purpose of this work was the synthesis of non-porous and nanocrystalline ScSZ layers on tubular substrates using MOCVD method at temperatures of 600–700 °C. As mentioned, ScSZ electrolytes are mainly manufactured by dip-coating method, and they are subsequently sintered at temperatures of 1400–1500 °C. The electrolytes obtained are often porous, especially when the sintering temperature is lower. Non-porous and nanocrystalline ScSZ layers have not been deposited on tubes at such low temperatures so far. So, this is a completely new approach to the problem. It is expected that the electrolytes synthesised using this technique could be useful for tubular SOFC operating at low temperatures. Reduction of the manufacturing and operating temperatures of ScSZ electrolyte would contribute to lower SOFC production costs and an increase in its efficiency, which is not possible using current techniques.

## 2. Materials and Methods

Scandia-stabilised zirconia layers were deposited on quartz glass substrates in the tube form with an internal diameter of 10 mm and a length of 25 mm. The layers were synthesised on the inner surfaces of tubes. The use of CVD or MOCVD allows to obtain non-porous and uniform in thickness layers even on substrates with complex shapes. In this work, tubular substrates were covered with ScSZ layers by MOCVD method using commercial Sc(tmhd)_3_ (Tris(2,2,6,6-tetramethyl-3,5-heptanedionato)scandium(III), 99%) from ABCR GmbH (Karlsruhe, Germany) and Zr(tmhd)_4_ (Tetrakis(2,2,6,6-tetramethyl-3,5-heptanedionato)zirconium)(IV), 99.99+%) from Sigma-Aldrich (Saint Louis, MO, USA) as basic reactants. The molar content of Sc(tmhd)_3_ in the gas mixture was as follows: 14 and 28%. The evaporation temperature of Sc(tmhd)_3_ was in the range of 130–150 °C, and in the case of Zr(tmhd)_4_ it was 250–270 °C. Ar (≥99.999% pure) was the carrier gas. Air was a source of oxygen as well as the carrier gas. Selected samples were obtained without the presence of air. The total gas pressure in the CVD reactor was changed from 5.5 kPa to 7.5 kPa. The layers were also deposited under atmospheric pressure. The synthesis temperature was 600–700 °C. The deposition time was in the range of 10–40 min. The synthesis process was carried out with low values of extended Gr_x_/Re_x_^2^ criterion [58], where Gr–Grashof number, Re- Reynolds number and x-the distance from the gas inflow point. It was assumed that its value should be less than 0.01. The substrate shape as well as its possible roughness were taken into account in establishing the value of this criterion. This expression is useful for synthesis of dense and uniform in thickness layers, because reactions in the gas phase (homogeneous nucleation) can be eliminated then. Formation of powders in the gas phase leads to the growth of porous layers with poor adhesion to the substrate [58,59]. This process also has a very significant influence on the layer thickness distribution on the substrate. The more intense homogeneous nucleation process, the greater differentiation in the layer thickness [60]. For these reasons, it is very important to eliminate this unfavourable process.

The diagram of the equipment for the layer deposition is demonstrated in the paper [61].

The layers obtained were examined using a scanning electron SEM NANO NOVA 200 from FEI EUROPE COMPANY (Eindhoven, The Netherlands) and an energy dispersive X-ray spectroscope (EDS) microanalyzer produced by EDAX EDS Company (Mahwah, NJ, USA). X-ray diffraction analyses were carried out with the use of an X’Pert X-ray diffractometer manufactured by Panalytical (Malvern, UK). The transparency tests of samples were performed using UV-VIS Spectrophotometer JASCO V630 from JASCO Deutschland GmbH (Pfungstadt, Germany).

## 3. Results and Discussion

Selected samples were subjected to microstructural and structural tests as well as chemical composition tests. Observation of the layer surfaces and their cross-sections performed with the use of scanning electron microscope indicates that the layers obtained under different conditions are without pores (Figure 1, Figure 2, Figure 3 and Figure 4, Figures 6 and 7). Figure 1 presents the surface of ZrO_2_-Sc_2_O_3_ layer deposited at temperature of 600 °C with 14 mol.% content of Sc(tmhd)_3_ in the gas mixture.

For comparison the microstructure of the layer obtained at the same temperature, but with 28 mol.% content of Sc(tmhd)_3_ is demonstrated in Figure 2. This layer also is without pores. The average EDS analyses from the layer surfaces shown in Figure 1 and Figure 2 confirm the presence of zirconium as well as scandium (Table 1 and Table 2).

Calculated molar contents of Sc_2_O_3_ in the layers are different. However, these values differ from those expected in both cases. When the molar content of Sc(tmhd)_3_ was 14% and the process was realised in argon and air at reduced pressure, the content of Sc_2_O_3_ in the layer should amount to 7%. Calculated value is 8.3% (Table 1). This value deviates slightly from the expected value.

For 28 mol.% content of Sc(tmhd)_3_ in reaction gas mixture it was expected that the molar content of Sc_2_O_3_ in the layer should be 14%. EDS analysis shows that it is 10.18% (Table 2). This means that dopant content in the layer is considerably lower. The differences in molar contents of reactants supplied to the CVD reactor and molar contents of reactants adsorbed on the substrate or the layer may be different when the synthesis process is realised in the regime controlled by the surface reaction rate. This takes place at low temperatures. Above the critical temperature designated as T^*^, the deposition process is controlled by the reactant diffusion to the substrate. At lower temperatures, competition in absorbance of reactants on the substrate is possible, while it does not occur at higher temperature [58]. For this reason, the expected content of reactants adsorbed on the substrate can be obtained. In addition, the synthesis process was realised without the presence of air. Oxygen included in air ensures elimination of carbon (solid by product of pyrolysis). Because the air was not present during the layer growth, the layer contained carbon. The presence of impurities also affects the surface reaction rate. Their presence also leads to reduction in the number of sites available for adsorption of the reactants. Furthermore, it should be noted that the layer was synthesised under a pressure close to atmospheric pressure when the concentration of reactants over the substrate is high. This factor also favours the occurrence of reactant competition in adsorption. A similar situation was observed in the work [52].

Figure 3 demonstrates the surface morphology (Figure 3a,b) as well as cross-section (Figure 3c) of the layer synthesised at a temperature of 650 °C with 14 mol.% content of Sc(tmhd)_3_ in argon and oxygen presence at near atmospheric pressure. The layer, as before, is non-porous. Its thickness is approx. 200 nm.

EDS analysis shows that the layer contains zirconium and scandium (Table 3). Sc_2_O_3_ content in the layer was 10.18 mol.% and it should be 7 mol.%.

In this case the synthesis process was also realised in the regime controlled by surface reaction rate. The deposition process was realised at near atmospheric pressure and although the process was carried out in presence of oxygen and carbon should not appear during the layer growth, the competition in the adsorption of reactants took place.

The microstructure and the cross-section of ZrO_2_-Sc_2_O_3_ layer also obtained at 650 °C, but with 28 mol.% content of Sc(tmhd)_3_ are shown in Figure 4. The layer was synthesised in the presence of argon and oxygen under reduced pressure. The synthesis time was two times longer and it was 40 min. The layer thickness also was two times thicker than previously. Its thickness was approx. 400 nm.

On the basis of X-ray analysis presented in Figure 5, it can be concluded that this layer is amorphous.

Results of EDS analysis indicate that the molar content of Sc_2_O_3_ in the layer was 14.09% (Table 4) which means that the value obtained is consistent with the expected value.

Probably then, the synthesis process was realised in the regime controlled by mass diffusion to the substrate. Reduction of the total gas pressure (without change of the partial reactant pressures) causes an increase in reactant diffusion to the substrate. In this way, the critical temperature T^*^ of transition from the process controlled by surface reaction rate to the process controlled by mass diffusion to the substrate can be lowered [58]. Then, the synthesis process can be realised in the regime controlled by reactant diffusion to the substrate and there is the correlation between chemical composition of reactants supplied to the CVD reactor and chemical composition of layers deposited.

Figure 6 demonstrates surface morphology (Figure 6a,b) and the cross-section: the layer (bright part)–the substrate (dark part) (Figure 6c) of ZrO_2_-Sc_2_O_3_ layer obtained at temperature of 700 °C at reduced pressure. The molar content of Sc(tmhd)_3_ in the gas reaction mixture was 14%. The synthesis time was only 10 min. The layer thickness was approx. 130 nm. (Figure 6c). Probably, numerous small crystallites (nanocrystallites) are visible on the surface (Figure 6a,b). A few aggregates consisting of small grains are also observed.

Table 5 contains results of the average EDS analysis. It was calculated that Sc_2_O_3_ content in this layer was 8.16 mol.%.

This value is close to the value obtained in the case of the sample presented in Figure 1. These layers were synthesised under similar conditions. They differed only in their deposition temperature. This can mean that the synthesis process was controlled by reactant diffusion to the substrate in both cases. Therefore, chemical composition of the layers can be controlled when their deposition process is conducted under above conditions.

Figure 7 presents the surface morphology (Figure 7a,b) and the cross-section: the layer (bright part)–the substrate (dark part) (Figure 7c) of the layer also synthesised at 700 °C, but with 28 mol% content of Sc(tmhd)_3_ in the gas mixture.

The synthesis time was 15 min., and the layer thickness was approx. 180 nm (Figure 7c). Small crystallites (nanocrystallites) are observed. The estimated grain size is approx. 25–40 nm. Several aggregates consisting of small grains are also present. However, they are more numerous and smaller than before. Results of X-ray analysis are demonstrated in Figure 8. Presented X-ray diffraction pattern shows broad peaks of low intensity, which indicates the initial formation of a crystalline phase with very fine grains (nanometre size). There are three diffraction peak positions. The first peak at a 2θ angle of 31.28 ± 0.06 degrees corresponds to a distance of 2.857 Å. The second peak at 2θ angle equal to 51.5 ± 0.4 degrees corresponds to a distance of 1.773 Å. A third peak at 2θ angle of 61 ± 1 degrees corresponds to a distance of 1.516 Å. Based on the position and intensity of the given peaks, the Zr_5_Sc_2_O_13_ compound with a rhombohedral structure was matched.

Data included in Table 6 indicate that the layer contains zirconium and scandium. The molar content of Sc_2_O_3_ in the layer is 13.97%.

Therefore, the Sc_2_O_3_ content in the layer obtained corresponds to the expected one. Furthermore, this value is close to the value obtained in the case of the layer presented in Figure 4 when the synthesis temperature was 650 °C and other conditions of the process were similar. Therefore, it can be concluded that the deposition process was also controlled by reactant diffusion to the substrate now.

Transparency of quartz glass tubes covered with ZrO_2_^−^Sc_2_O_3_ layers under different conditions and uncoated glass was tested using UV-Vis spectrophotometer. Results of these tests are demonstrated in Figure 9.

It can be concluded that the synthesis process under reduced (curves 1, 4, 5 and 6) as well as near atmospheric pressure (curves 2 and 3) was realised without the presence of homogeneous nucleation. Its presence could considerably reduce the glass transparency, because the layers would be characterised by high porosity then and consequently light would be scattered on the pores.

In the case of quartz glass with ZrO_2_-Sc_2_O_3_ layers deposited at temperature of 700 °C (curves 5 and 6), the transparency is the most reduced. This is due to the fact that the layers obtained at this temperature were already nanocrystalline and the light was scattered at grain boundaries. Furthermore, in the case of curve 3, a reduction of the layer transparency in the wavelength range of 900–1100 nm is also observed. It may have been caused by the roughness of the commercial substrate.

It was expected that the presence of carbon in the layer (curve 2) should significantly reduce the sample transparency, but, however, it is only slightly lowered. Probably, the layer was thin and contained small amounts of carbon.

## 4. Conclusions

Non-porous amorphous or nanocrystalline ZrO_2_-Sc_2_O_3_ layers were deposited on tubular substrates via MOCVD method using Zr(tmhd)_4_ and Sc(tmhd)_3_ at temperatures of 600–700 °C, significantly lower than temperatures used in other techniques. Argon and oxygen were used as carrier gases. Air also was a source of oxygen. The content of Sc(tmhd)_3_ in reaction gas mixture was 14 and 28 mol.%. When the layers were synthesised under reduced pressure there was a correlation between chemical compositions of reactants supplied to the CVD reactor and the layers obtained. It is possible when the deposition process is realised in the regime controlled by reactant diffusion to the substrate. At near atmospheric pressure there was no such correlation. The synthesis process was carried out in the regime controlled by surface reaction rate. ZrO_2_-Sc_2_O_3_ layers deposited at a temperature of 700 °C were nanocrystalline. Sc_2_O_3_-stabilised ZrO_2_ containing approx. 14 mol.% Sc_2_O_3_ had a rhombohedral structure.

## Figures and Tables

**Figure 1 materials-15-02120-f001:**
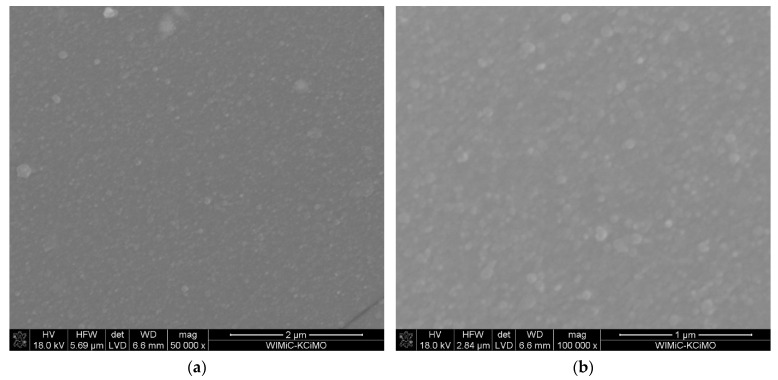
Microstructure of the layer deposited at temperature of 600 °C on tubular substrate at different magnifications: 50,000× (**a)** and 100,000× (**b**). The content of Sc(tmhd)_3_ in the gas mixture: 14 mol.%. Total gas pressure: 6580 Pa. Synthesis time: 20 min.

**Figure 2 materials-15-02120-f002:**
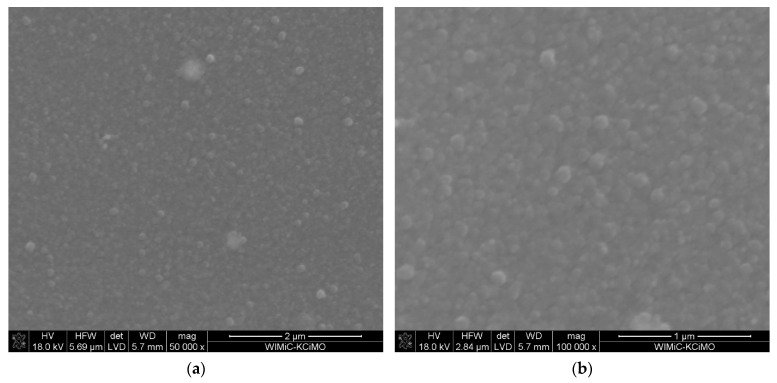
Microstructure of the layer deposited at temperature of 600 °C on tubular substrate without the presence of air at different magnification: 50,000× (**a**) and 100,000× (**b**). The content of Sc(tmhd)_3_ in the gas mixture: 28 mol.%. Total gas pressure: 10^5^ Pa. Synthesis time: 20 min.

**Figure 3 materials-15-02120-f003:**
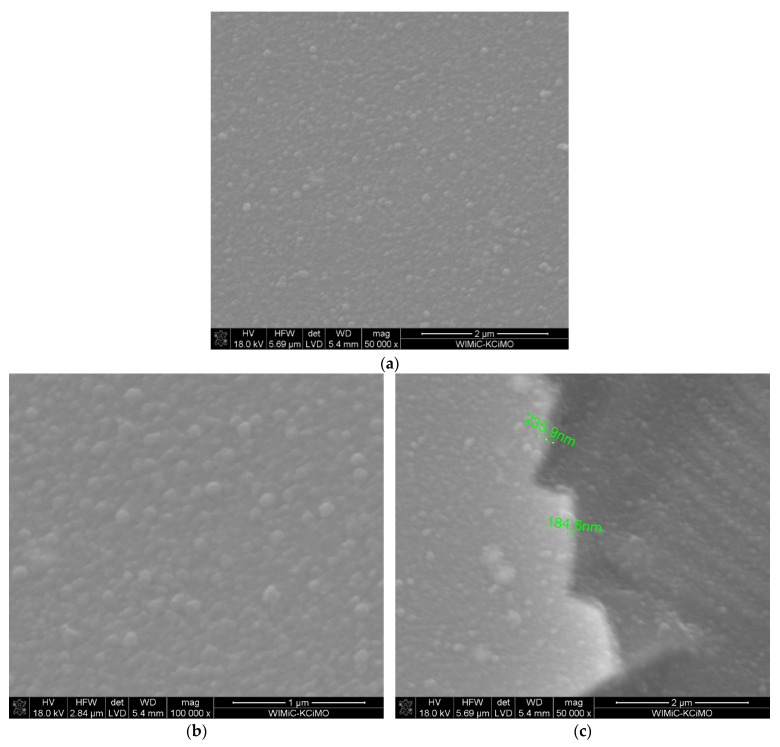
Microstructure of the layer synthesised at a temperature of 650 °C on tubular substrate at different magnifications: 50,000× (**a**) and 100,000× (**b**) and the cross-section: the layer–the substrate (**c**). The content of Sc(tmhd)_3_ in the gas mixture: 14 mol.%. Total gas pressure: 10^5^ Pa. Deposition time: 20 min.

**Figure 4 materials-15-02120-f004:**
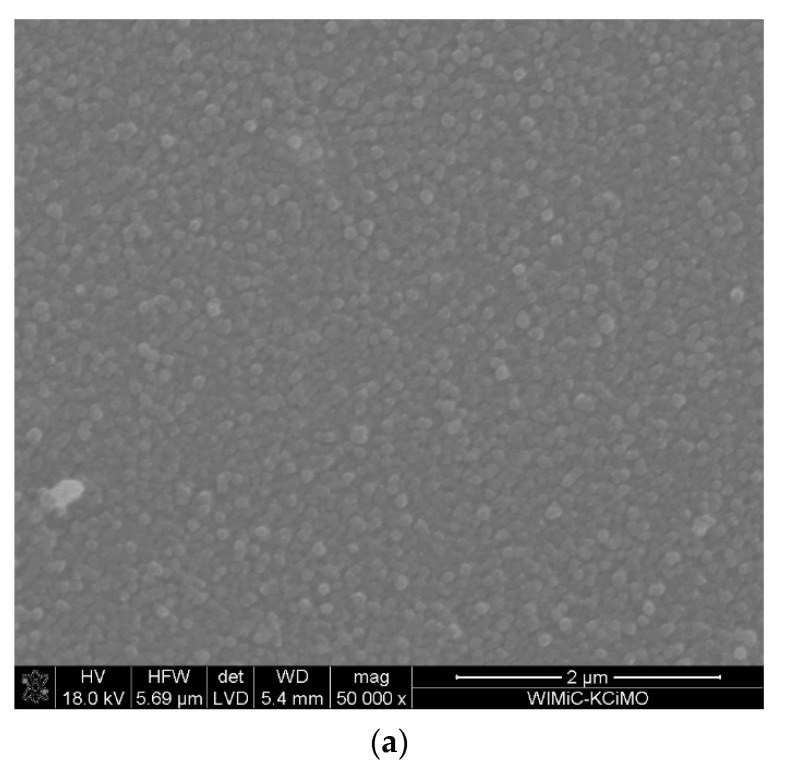
Microstructure of the layer obtained at temperature of 650 °C on tubular substrate at different magnifications: 50,000× (**a**) and 100,000× (**b**) and the cross-section: the layer (bright part)–the substrate (dark part) (**c**). The content of Sc(tmhd)_3_ in the gas mixture: 28 mol.%. Total gas pressure: 6580 Pa. Deposition time: 40 min.

**Figure 5 materials-15-02120-f005:**
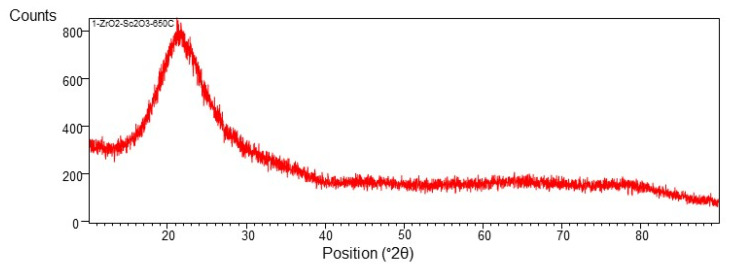
X-ray diffractogram of the layer deposited at 650 °C (demonstrated in Figure 4).

**Figure 6 materials-15-02120-f006:**
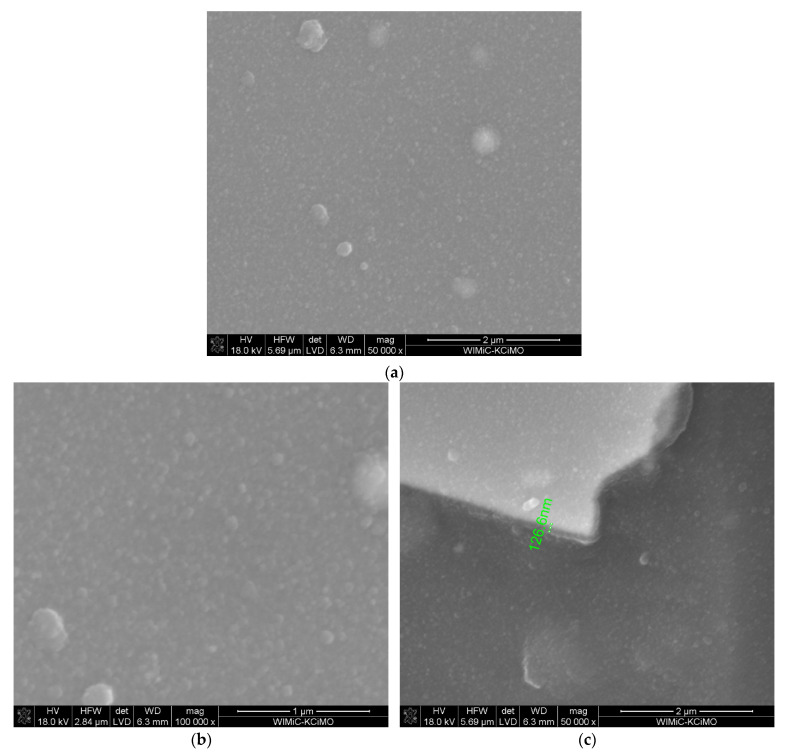
Microstructure of the layer synthesised at temperature of 700 °C on tubular substrate at different magnifications: 50,000× (**a**) and 100,000× (**b**) and the cross-section: the layer–the substrate (**c**). The content of Sc(tmhd)_3_ in the gas mixture: 14 mol.%. Total gas pressure: 6580 Pa. Deposition time: 10 min.

**Figure 7 materials-15-02120-f007:**
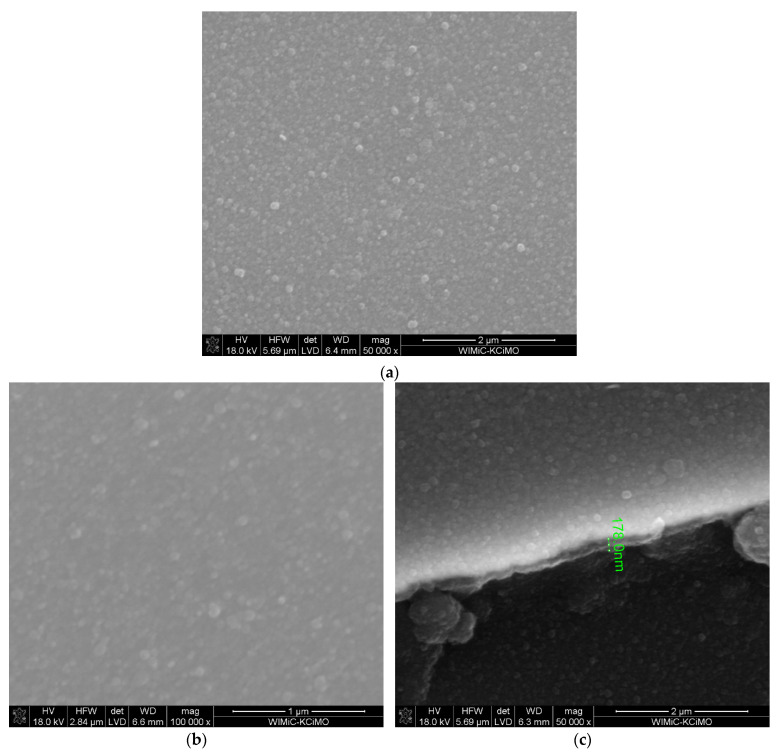
Microstructure of the layer deposited at temperature of 700 °C on tubular substrate at different magnifications: 50,000× (**a**) and 100,000× (**b**) and the cross-section: the layer–the substrate (**c**). The content of Sc(tmhd)_3_ in the gas mixture: 28 mol.%. Total gas pressure: 6580 Pa. Synthesis time: 15 min.

**Figure 8 materials-15-02120-f008:**
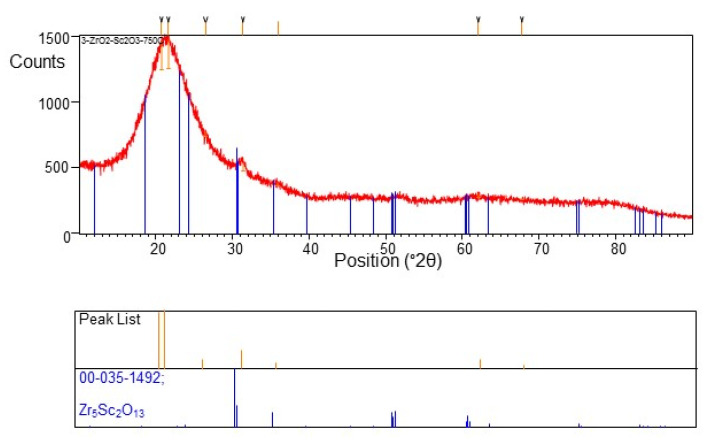
X-ray diffractogram of the layer obtained at 700 °C (presented in Figure 7).

**Figure 9 materials-15-02120-f009:**
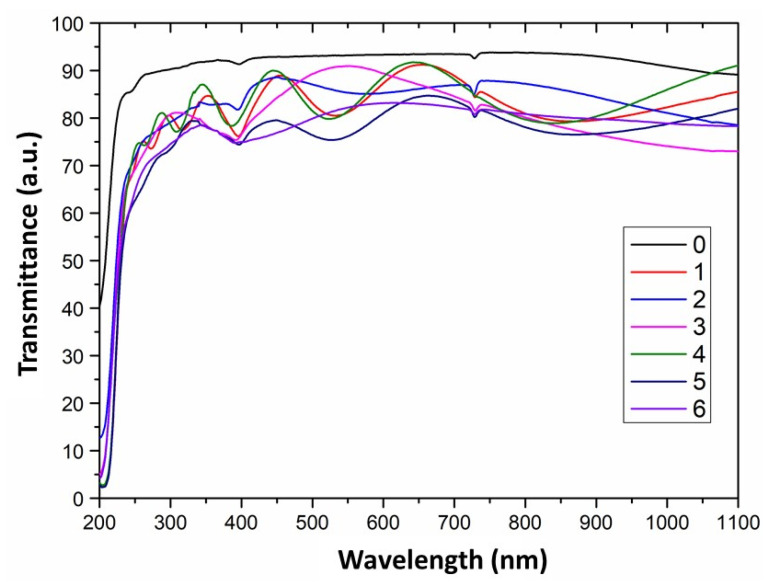
Transmittance spectrum of uncoated quartz glass and covered with ZrO_2_-Sc_2_O_3_ layers under various conditions. 0—uncoated quartz glass; 1—with ZrO_2_-Sc_2_O_3_ layer synthesised at 600 °C (the content of Sc(tmhd)_3_: 14 mol.%), reduced pressure; 2—with ZrO_2_-Sc_2_O_3_ layer synthesised at 600 °C (the content of Sc(tmhd)_3_: 28 mol.%) without air presence, atmospheric pressure; 3—with ZrO_2_-Sc_2_O_3_ layer synthesised at 650 °C (the content of Sc(tmhd)_3_: 14 mol.%), atmospheric pressure; 4—with ZrO_2_-Sc_2_O_3_ layer synthesised at 650 °C (the content of Sc(tmhd)_3_: 28 mol.%), reduced pressure; 5—with ZrO_2_-Sc_2_O_3_ layer synthesised at 700 °C (the content of Sc(tmhd)_3_: 14 mol.%), reduced pressure; 6—with ZrO_2_-Sc_2_O_3_ layer synthesised at 700 °C (the content of Sc(tmhd)_3_: 28 mol.%), reduced pressure.

**Table 1 materials-15-02120-t001:** Average chemical composition determined by EDS analysis and calculated molar content of Sc_2_O_3_ in the layer (Figure 1). The synthesis temperature: 600 °C. Total gas pressure: 6580 Pa. The molar content of Sc(tmhd)_3_ in the gas mixture: 14%. The layer was deposited without the presence of air.

Element	Atomic [%]	The Molar Content of Sc_2_O_3_ in the Layer [%]
Zr	10.43	8.30
Sc	1.89	

**Table 2 materials-15-02120-t002:** Average chemical composition determined by EDS analysis and calculated molar content of Sc_2_O_3_ in the layer (Figure 2). The synthesis temperature: 600 °C. Total gas pressure: 10^5^ Pa. The molar content of Sc(tmhd)_3_ in the gas mixture: 28%.

Element	Atomic [%]	The Molar Content of Sc_2_O_3_ in the Layer [%]
Zr	11.51	10.18
Sc	2.61	

**Table 3 materials-15-02120-t003:** Average chemical composition determined by EDS analysis and calculated molar content of Sc_2_O_3_ in the layer (Figure 3). The synthesis temperature: 650 °C. The molar content of Sc(tmhd)_3_ in the gas mixture: 14%. Total gas pressure: 10^5^ Pa.

Element	Atomic [%]	The Molar Content of Sc_2_O_3_ in the Layer [%]
Zr	13.81	10.18
Sc	3.14	

**Table 4 materials-15-02120-t004:** Average chemical composition determined by EDS analysis and calculated molar content of Sc_2_O_3_ in the layer (Figure 4). The synthesis temperature: 650 °C. Total gas pressure: 6580 Pa. The molar content of Sc(tmhd)_3_ in the gas mixture: 28%.

Element	Atomic [%]	The Molar Content of Sc_2_O_3_ in the Layer [%]
Zr	12.98	14.09
Sc	4.26	

**Table 5 materials-15-02120-t005:** Average chemical composition determined by EDS analysis and calculated molar content of Sc_2_O_3_ in the layer (Figure 6). The synthesis temperature: 700 °C. Total gas pressure: 6580 Pa. The molar content of Sc(tmhd)_3_ in the gas mixture: 14%.

Element	Atomic [%]	The Molar Content of Sc_2_O_3_ in the Layer [%]
Zr	20.83	8.16
Sc	3.70	

**Table 6 materials-15-02120-t006:** Average chemical composition determined by EDS analysis and calculated molar content of Sc_2_O_3_ in the layer (Figure 7). The synthesis temperature: 700 °C. Total gas pressure: 6580 Pa. The molar content of Sc(tmhd)_3_ in the gas mixture: 28%.

Element	Atomic [%]	The Molar Content of Sc_2_O_3_ in the Layer [%]
Zr	13.33	13.97
Sc	4.33	

## Data Availability

The data presented in this study is available on request from the corresponding author.

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
