# Peer review of "Chemical Vapour Deposition of Scandia-Stabilised Zirconia Layers on Tubular Substrates at Low Temperatures"

_materials, 2022, doi:10.3390/ma15062120_

Round 1
Reviewer 1 Report
The author studied scandia-stabilized zirconia layers on tubular substrates by MOCVD at low temperature, the results are interesting, however, something must be improved.
- Sc2O3-stabilized ZrO2 layers were deposited on quartz glass substrates, how to ensure its uniform deposition or do you have any figures that can prove it?
- The figures about surface topograph were a bit blurry, as a result, some key and important information cannot be clearly reflected. The measurement of the thickness existed errors.
- It is not enough to claim that the layers were good without pores, how about the inside of the layers?
- The author calculated the molar content of Sc2O3 in the layer basing on the results of EDS. The results of EDS only proved the existence of the elements, it may be difficult to explain the form in whick the elements existed in the layer.
- The introduction was detailed, however, the author should explain the question about the deposition on the tubular to highlight the innovation of the paper. Is teh method new? Is the formula preparing the layers new? Or is there any difficulties that hinder the deposition of the layers on the tubular before?
- The conclusion about the nanocrystalline layers was not convincing just by the observation of the surface topography.
- The characteristic peaks in Fig. 8 were not obvious, so the conclusion that "They confirm that the layer is crystalline. Scandia-stablized zirconia obtained has a rhombohedral structure. " was not so convincing.
Reviewer 2 Report
Manuscript ID materials-1630859
Full Title: Chemical vapour deposition of scandia-stabilized zirconia layers on tubular substrates at low temperatures.
Journal of Materials
Dear Editor,
The manuscript, entitled « Chemical vapour deposition of scandia-stabilized zirconia layers on tubular substrates at low temperatures » . presents finding of study on synthesis of non-porous ZrO2-Sc2O2 layers 8 on tubular substrates by Metalorganic Chemical Vapour Deposition(MOCVD) method using 9 Sc(tmhd)3 (Tris(2,2,6,6-tetramethyl-3,5-heptanedionato)scandium(III), 99%) and Zr(tmhd)4 10 (Tetrakis(2,2,6,6-tetramethyl-3,5-heptanedionato)zirconium)(IV), 99,99+% ) as basic reactants.
This study is a good paper and deals with an interesting subject of the synthesis of non-porous and nanocrystalline ScSZ layers on tubular substrates using MOCVD method at temperatures of 600-700℃.
Nevertheless, I have some comments on it, as follows:
1) It is recommended the authors try to explain the novelty of the paper as clear as possible and explain the research gap they are trying to fill.
2) It is necessary for authors to develop the introduction and to explain the improvement they have achieved compared to previously published results.
3) The References should be updated (2019 to 2022).
4)The authors should give the advantage of the chemical vapour deposition method in the introduction.
In summary, this study provides detailed responses so this work can be interesting for further incoming research. This article can be accepted for publication in materials if the authors provide the required amendments of the above-mentioned comments convincingly.
Author Response
Pleas see the attachment

Reviewer 3 Report
In the manuscript, “Chemical vapour deposition of scandia-stabilized zirconia layers on tubular substrates at low temperatures,” the author varied the temperature and atmosphere to grow different ScSZ layers on tubular substrates. However, the quality of obtained ScSZ and its capability for SOFC application is unknown. Here are my comments:
- As discussed in the introduction part, conductivity and mechanical strength are two key factors for synthesized ScSZ solid oxides. The author should at least measure the conductivity of MOCVD-obtained ScSZ. Otherwise, readers can’t know whether it is a good method to prepare ScSZ or not.
- For Figures 1 and 2, the author mentioned that their microstructures are similar. However, the grain sizes look significantly different. The author can provide the estimated grain size for different ScSZ layers for readers.
- The cross-sectional SEM in Figures 3, 4, 6 look strange and it’s hard to identify the materials (what are the bright and dark parts?) maybe because they are in tubular shapes. The author can put SEM images with smaller magnification as insets to help identify the materials and cross-section.
Round 2
Reviewer 1 Report
The results are good.
Reviewer 3 Report
It's a pity that the authors cannot include conductivity information although the measurement is quite simple. I don't have other questions.